# Effects of the Dietary Inclusion of Buriti Oil on Lamb Performance, Carcass Traits, Digestibility, Nitrogen Balance, Ingestive Behavior and Blood Metabolites

**DOI:** 10.3390/ani10111973

**Published:** 2020-10-28

**Authors:** Luciana Diogénes, Leilson Bezerra, José Pereira Filho, Jarbas Silva Junior, Juliana Oliveira, José Moura, Analivia Barbosa, Mateus Souza, Sheila Sousa, Elzânia Pereira, Ronaldo Oliveira

**Affiliations:** 1Center of Health and Agricultural Technology, Federal University of Campina Grande, Avenida Universitária, s/n-Jatobá, Patos PB 58708110, Patos, Paraíba, Brazil; luhvianadiogenes@hotmail.com (L.D.); jmorais@cstr.ufcg.edu.br (J.P.F.); jupaula.oliv@yahoo.com.br (J.O.); jose.fabio@ufcg.edu.br (J.M.); 2Department of Animal Science, Federal University of Bahia, Av. Adhemar de Barros, 500, Ondina, Salvador 40170110, Bahia, Brazil; miguelreges@gmail.com (J.S.J.); analiviabarbosa@gmail.com (A.B.); mateusnetosilva@hotmail.com (M.S.); 3Department of Animal Science, Federal University of Piaui, Ininga, S/N, Teresina 64049-550, Piauí, Brazil; sheila_vilarindo@hotmail.com; 4Department of Animal Science, Federal University of Ceara, 2977, Mister Hull Avenue, Fortaleza 60356000, Ceara, Brazil; elzania@hotmail.com

**Keywords:** byproduct, carcass, lipid supplementation, *Mauritia flexuosa* L., sheep

## Abstract

**Simple Summary:**

Dietary fat is important for animals, not only because it supplies essential fatty acids and fat-soluble vitamins, but also because of its high energy content, approximately twice as many calories per gram of carbohydrates. Thus, it induces a caloric increase, in addition to modulating the fatty acid (FA) profile of meat. Vegetable oils inclusion in the diet of lambs improves the lipid quality of meat. Among them, buriti oil (*Mauritia flexuosa* L.F.) is an example that stands out; it can represent an important alternative to meet the energy demands of lambs, particularly because of its easy availability, especially in the northern and northeast states of Brazil. In addition, as a consequence of the high cost of protein supplements in feed concentrates, unconventional alternatives have been exploited in recent years. Thus, this research proposes the use of a byproduct of the biofuel industry. This study was conducted to enhance our knowledge of interesting opportunities for farmers in terms of lamb meat production. Its use could promote activity in the livestock sector by reducing feed costs and becoming an alternative to producers without easy access to more expensive supplements.

**Abstract:**

Buriti (*Mauritia flexuosa* L.) oil (BO) is a byproduct that can be incorporated into the diet of lambs, thus increasing the energy density. The study aimed to evaluate the optimum BO inclusion level in lamb diets. Sixty-five Santa Ines lambs were distributed in two completely randomized experiments with five treatments each (BO inclusion at 0 (control), 12, 24, 36 and 48 g/kg dry matter (DM) total). The BO supplementation to partially replace ground corn linearly decreased the nutrient intake and digestibility of DM, ether extract and neutral detergent fiber (NDF), N° chews/bolus, DM and NDF rumination or eating efficiencies, the N intake and N balance, carcass weights and yields, and dressing content of lamb carcasses (*p* < 0.05). The addition of BO responded quadratically to DM eating efficiency and N-fecal and N-urinary excretion (*p* < 0.05). The linear response plateau (LRP) analysis demonstrated that the levels of 0 and 12 g/kg of BO were similar, and improved intake and digestibility and consequently performance (*p* < 0.001). There was a linear increase in feed efficiency and shrinkage after chilling with the BO inclusion replacing ground corn. The inclusion of 12 g/kg BO in the total DM of diet is recommended, because it improves feeding intake, digestibility and performance of lambs.

## 1. Introduction

In animal feed, researchers are studying alternative sources of feed, such as the byproducts derived from biodiesel, which can reduce production system costs and improve the profitability of producers without harming animal performance [1,2,3]. Buriti oil (BO) has significant concentrations of lipids [4], and it may be used in animal feed as an energy ingredient for total or partial replacement of ground corn [4,5,6], which is commonly used in animal feed [7,8,9].

The great benefit of using vegetable seed oils such as BO is the dietetic energy increase, as it has greater energy content than carbohydrate sources, increasing the energy density of diets, reducing fermentation, caloric increment, and improving productive efficiency [10]. However, vegetable oils are considered highly unsaturated sources and can therefore alter the metabolism of the microbial population in the rumen and, consequently, intake, digestibility, and ingestive behavior of ruminants [11,12]. Zinn and Jorquera [13] affirm that, regardless of the source and form of fat addition to the diet, the nutrient value of fat assigned by the National Research Council-NRC [14] tends to be consistent when total fat intake does not exceed the proportion of 0.96 g intake fat/kg body weight (BW). If fat intake is higher than this proportion, the energy value of fat decreases linearly as a direct result of the reduction of intestinal fatty acid (FA) digestibility (mainly C18:0), possibly because of a limited bile production capacity [14]. Intestinal FA digestibility, and the NE value of supplemental fats used in feedlot diets, is a highly predictable function based on the total FA intake per unit of BW. In contrast, energetic supplements with using oils can reduce costs of diets if precautions are taken to maintain the balance of ruminal energy/protein [4,15], in addition to correcting unbalanced diets, increasing feeding efficiency and also meat production [2].

Buriti oil is a byproduct from the mechanical pressing and chemical extraction of “Buritizeiro” palm seeds (*Mauritia flexuosa* L.), and has great potential to be used as a supplement in ruminant diets; its use has been investigated in studies on production of milk [4] and meat, as well as in manufactured products [16,17]. BO extracted from the pulp of the fruit of buriti has in its composition carotenoids, tocopherols, and fatty acids consisting mostly of fatty acids oleic (73%) and palmitic (16:0; 31%) [4,16,17]. According Parente et al. [17], BO can be an alternative to replace conventional lambs diet ingredients, in order to meet the energy demand of the animals, as well as to improve meat quality, increasing functional compounds beneficial to consumers’ health. However, it is important to consider the addition at moderate concentrations due to lipids toxic effects on ruminal bacteria, their use in diets [4,16,17].

Thus, considering the availability and nutritional characteristics, it was hypothesized that diets with a BO inclusion up to the maximum level of 48 g/kg dry matter (DM) can be used for confined Santa Ines lambs, contributing to an energetic increase in the diet without harming rumen bacteria, due to the toxic effect of unsaturated fatty acids (UFA). Therefore, we aimed to evaluate the optimum levels of BO inclusion on performance, carcass traits, digestibility, ingestive behavior, and blood serum metabolites of Santa Ines lambs.

## 2. Materials and Methods

Two experiments were conducted with a total of sixty-five uncastrated Santa Ines lambs at the Federal University of Bahia, Salvador, Bahia, Brazil, and all management practices followed the recommendations of the National Council for the Control of Animal Experimentation (Permit Number: 306-17), in accordance with the precepts of Law No. 11794, of 8 October 2008, and the rules issued by the National Council of Animal Experimentation Control (CONCEA, Brazil).

### 2.1. Experiment 1: Animals and Experimental Design

The experiment lasted 70 days, and was preceded by a 14-day adaptation period. Forty uncastrated Santa Ines lambs with an average age of four months and a mean BW of 28.0 ± 0.5 kg were treated against internal and external parasites with ivermectin (Ivomecgold^®^, Merial, Salvador, Bahia, Brazil) and against clostridiosis with a polyvalent vaccine (Sintoxan^®^, Merial, Salvador, Bahia, Brazil) during the adaptation period. Lambs were distributed in a completely randomized design with five treatments and eight experimental units.

#### 2.1.1. Experimental Diets and Management

The animals were housed in covered sheds containing 40 pens with a concrete floor (2.0 × 2.0 m) and 25 metabolism test cages (0.75 × 1.2 m) provided with drinking fountains and feeders. Diets were formulated according to NRC [14]; they were isonitrogenous and were formulated for male lambs for an average daily gain (ADG) of 200 g. The concentrate feed (400 g/kg DM total) was combined with 0 g/kg, 12 g/kg, 24 g/kg, 36 g/kg, or 48 g/kg DM BO (handmade by local producers; diets were formulated to contain BO and to be isonitrogenous). Corn was coarsely ground using a grinder (Trapp, TRF 80, Jaraguá do Sul, SC, Brazil), and soybean meal and a mineral mixture were also included in the diets. BO was obtained by double pressing whole grains using a mechanical press-type “expeller” (Pelegrin^®^, Thermal Press, Sao Paulo, Brazil).

The determination of fatty acid (FA) profile was carried out from the fatty acid methyl esters (FAME), which were previously extracted from ingredients of the diets (Table 1), conducted according to the method described by O’Fallon et al. [18], in which a solution of potassium hydroxide, methanol, sulfuric acid, hexane and internal standard C19:0 was used. Transmethylated samples of ingredients were analyzed in a gas chromatograph (Finnigan Focus GC, Varian, CA, USA) with a flame ionization detector and a capillary column (SLB-ILL111, 100 m long by 0.25 mm internal diameter and 0.20 µm film thickness, Sigma-Aldrich).

BO was added to the concentrate in the diet immediately before feeding. The Tifton-85 hay used as roughage (600 g/kg DM total) was separately weighed using an electronic scale (Welmy, BCW 6/15/30, Santa Bárbara d’Oeste, SP, Brazil), and concentrate and roughage were mixed and offered daily. The amount of feed administered twice daily at 7:30 and 16:30 h, and refusal was recorded daily to adjust the feed offered for 100 g/kg DM refusal. Both feed and refusals were sampled weekly and frozen at −20 °C for further analysis. The chemical composition of dietary ingredients is shown in Table 1, while the ingredient proportion and chemical composition of diets are described in Table 2.

#### 2.1.2. Intake, Performance and Carcass Traits

Nutrient intake was estimated by the difference between the total concentration of each nutrient in the feed offered to the lambs and the amount of nutrient in refusals. Animals were weighed using an electronic scale (Welmy, W 300, Salvador, Bahia, Brazil) after a 16 h fast, on the first day (initial BW), every 20 days (ADG) and at the end of the 70-day feeding trial (final BW). Then, the animals were subjected to solid fasting and free water for 14 h and weighed to determine the slaughter body weight (SBW). The slaughter was carried out in a commercial slaughterhouse following the directives of the Brazilian Federal Inspection Service (BFIS). The lambs were then bled, skinned, and eviscerated, and the hot carcass weight (HCW) was obtained. After 24 h of cooling at 4 °C, the carcasses were weighed to determine cold carcass weight (CCW). The dressing percentage (DP) and shrinkage after chilling (SC) were calculated by the following formulae [19]:DP = (HCW/SBW) × 100 and SC = ([HCW − CCW]/HCW) × 100.(1)

Subcutaneous fat thickness (SFT, mm) was measured in the *longissimus* muscle (LM) area (LMA, cm^2^) between the 12th and 13th ribs [19]. After 24 h of cooling, the LM was transversely cut, and the-SFT was measured from both sides of the carcass using an outside digital caliper (Digimess^®^, São Paulo, SP, Brazil). The exposed side of the LM was measured with a plastic grid to obtain the LMA.

#### 2.1.3. Ingestive Behavior

Observations of the 40 lambs were performed individually on the 49th day at five-minute intervals for 24 h to evaluate ingestive behavior according to Martin and Bateson [20]. The behavioral activities (eating, ruminating, and time spent idling) were recorded by two trained observers, who were positioned in places where they could interfere as minimally as possible with the animal behavior.

The chews performed during the rumination were evaluated during three rumination periods at three different times of the day (10–12 h, 14–16 h, and 18–20 h). The number of ruminating chews and the time spent ruminating each bolus (seconds/bolus) were determined using a digital chronometer [20]. Chewing was calculated over three 15 s periods by multiplying the average number of chews by four to obtain the chewing time per minute (chew/min). The eating efficiency and rumination efficiency (RE) of DM and NDF and total chewing time (TCT, h/day), being the sum of the time spent eating and ruminating, were calculated according to Bürger et al. [21].

### 2.2. Experiment 2: Digestibility Trial, N Balance and Blood Metabolites

In the second experiment, twenty-five lambs (five experimental units) with an average BW of 28.0 ± 0.5 kg were distributed in metabolic cages during the first 28 days of the feedlot trial receiving the same diets and treatments and undergoing the same management of this diet for digestibility and N-balance determination. The feces, urine, and refusals from each animal were collected and quantified during this period (total collection). Each metabolic cage was equipped with a device for separation and total collection of feces and total collection of urine. After a 21-day adaptation period to the diet and cage, two daily fecal collections at 8:00 and 16:00 h were performed for seven consecutive days.

Every day, an aliquot (30%) of the total excreted feces was sampled and stored in plastic bags. To avoid the volatilization of nitrogen compounds from urine, a 10 N hydrochloric acid solution was placed in the container before sampling at a volume within 10 mL of the amount of urine produced from the previous day. Composite samples of feces and refusals of each animal were prepared throughout the collection period and stored at −20 °C for further analysis.

To calculate DM, crude protein (CP), NDF, and ether extract (EE) digestibility coefficients (DCs), the following equation was used: DC = [(kg of the portion ingested-kg of the portion excreted)/(kg of the portion ingested)] × 100. The intake of total digestible nutrients (TDN) was calculated according to suggestions of Sniffen et al. [22], using the equation ITDN = [(ICP − CPf) + 2.25 (IEE − EEf) + ITC − TCf], where ICP, IEE, and ITC represent the intake of CP, EE, and total carbohydrates (TC), respectively, and where CPf, EEf, and TCf refer to the excretion of CP, EE, and TC in feces, respectively. Concentrations of dietetic TDN were calculated with the equation TDN = (TDN intake/DM intake) × 100.

The nitrogen (N) contents of triplicate samples of the provided diet, feces and urine were determined according to AOAC [23] method 981.10. The balance (retention) of N (N retained, g/d) was obtained using N balance (g/d) = N intake (g/d) − [N excreted in feces (g/d) + N excreted in urine (g/d)]. BEN (basal endogenous nitrogen) was obtained using the equation BEN (g/d) = (0.018 + 0.35) × BW^0.75^. BEN was considered a loss of endogenous tissue and dermal N to be 0.35 and 0.018 in metabolic weight, respectively [24].

On the 4th and 6th days of the experimental trial, blood samples were collected by jugular vein puncture in vacutainer tubes (Labtest^®^ Diagnóstico SA, Minas Gerais, Belo Horizonte, Brazil) in the morning, immediately after feeding, and after 4 h. Disposable needles (25 mm × 8 mm) were used, and 10-mL blood samples were placed in glass tubes without anticoagulant for biochemical tests. Metabolic parameters were analyzed according to Labtest^®^ methods (Diagnostic SA, Minas Gerais, Brazil): albumin was measured by bromocresol green; total protein concentration was measured by the biuret method; total cholesterol was measured by the cholesterol enzymatic method; the globulin concentration was calculated by the mathematical difference between total protein and the albumin serum concentration. The albumin:globulin (A:G) ratio was also calculated. The laboratory analyses were conducted via the colorimetric method on a semiautomatic biochemical analyzer (BIOPLUS 2000^®^, São Paulo, Brazil).

### 2.3. Chemical Analysis and Calculations

The samples of ingredients, refusals, and feces were then thawed, dried in a forced-air oven at 55 °C for 72 h, ground using a Wiley cutting mill with a 1 mm mesh sieve, and then analyzed in triplicate to determine DM (method 967.03), ash (method 942.05), CP (method 981.10), and EE (method 920.29) content [22]. Neutral detergent fiber (NDF) and acid detergent fiber (ADF) were determined according to Van Soest et al. [25], with the modifications proposed in the Ankom device manual (Ankom 200, Technology Corporation, Macedon, New York, NY, US). NDF residue was incinerated in an oven at 600 °C for 4 h, and protein correction was determined by subtracting the neutral detergent insoluble protein (NDIP). Acid detergent lignin (ADL) content was determined from the ADF residue treated with 72% sulfuric acid. From ADL, the other cell wall fractions were estimated according to the equations described as hemicellulose = NDF − ADF and cellulose = ADF − ADL [24]. Nonfibrous carbohydrate (NFC) content was calculated according to Mertens [26], using the value of NDF corrected for ash and protein. NDIP and acid detergent insoluble protein (ADIP) contents were determined according to Licitra et al. [27]. The Small Ruminant Nutrition System, v. 1.8.6 [28] was used to calculate the metabolizable energy (ME) of the diets.

### 2.4. Statistical Analysis

The experimental design of the first and second experiments was completely randomized with five treatments with eight and five experimental units per treatment, respectively. The following statistical model was used:Yij = μ + si +eij,(2)
where Yij = the observed value, μ = the overall mean, si = the effect of BO (0 for the control or 12, 24, 36 and 48 g/kg of total DM), and eij = the effect of the experimental error. A general linear model was used to perform a linear regression using the PROC GLM from SAS^®^ [29].

To contribute to data homogeneity, when analyzing the growing performance and carcass trait data, the initial BW was used as a covariate for statistical analysis using the following model:Yij *=* µ *+* Ti *+* β(Wij − W) *+* eij,(3)
where Yij = the observed value of the dependent variable (performance and carcass) in animal j receiving treatment i; μ = the general mean; Ti = the fixed treatment effect i (i = the effect of the BO inclusion); β = the linear regression coefficient relative to covariate Wij; Wij = the covariate effect (initial BW of animal j receiving treatment i); and eij = random effects in the experimental error.

For the other data, the following statistical model was used:Yij = μ + si +eij,(4)
where Yij = the observed value, μ = the general mean, si = the effect of the level of the BO, and eij = the effect of the experimental error in the plots.

To assess what BO inclusion had the highest influence on the data, the linear response plateau (LRP) was analyzed using the PROC NLIN command (from SAS 9.4^®^ statistical software). In addition, to determine the relationship between the rate of inclusion and each evaluated parameter and to find the best rate of BO addition, a polynomial contrast was used to determine the linear and quadratic effects of treatments. The PROC NLIN software SAS 9.4^®^ command was also used for LRP analysis because of the biological phenomena presented by the ingestive behavior variables. Significance was declared when *p* ≤ 0.05.

## 3. Results

### 3.1. Intake, Performance and Carcass Traits

BO inclusion in feed concentrate to partially replace ground corn decreased linearly the intake of DM (*p* = 0.003), CP (*p* = 0.011), EE (*p* = 0.006), NFC (*p* = 0.017) TDN (*p* = 0.011), and ME (*p* < 0.001) but merely caused a trend-level linear reduction in NDF_ap_ (*p* = 0.088) intake by the lambs (Table 3).

The partial replacement of ground corn with BO in the feed concentrate promoted a linear decrease in the HCW (*p* = 0.009), CCW (*p* = 0.009), HCY (Hot carcass yield) (*p* = 0.001), CCY (Cold carcass yield) (*p* = 0.002), dressing content (*p* = 0.035) and SC (*p* = 0.043) of lamb carcasses. However, BO inclusion in lamb diet did not change the total weight gain (TWG; *p* = 0.13), ADG (*p* = 0.13), slaughter BW (*p* < 0.001), LMA (*p* = 0.14), or SFT (*p* = 0.91). There was a linear increase (*p* = 0.012) in feed efficiency (g ADG: g DMI ratio), with the inclusion of up to 48 g/kg BO in the feed concentrate as a replacement for ground corn. From the LRP test, it is observed that the inclusion of 12 g/kg of BO in DM improved performance variables and carcass traits (*p* < 0.001), where we highlight attention to the ADG, carcass weight and yields and LMA that contribute significantly to the choice of diet and the best performance of animals.

### 3.2. Ingestive Behavior

The different BO levels added to the concentrate did not affect (*p* > 0.05) the time spent eating (*p* = 0.51), chewing (*p* = 0.75), ruminating (*p* = 0.17), or idling (*p* = 0.76). However, the number of chews/bolus (*p* = 0.014), DM eating efficiency (*p* = 0.031), DM (*p* = 0.0002), and NDF (*p* = 0.019) RE reduced linearly with increasing BO inclusion in the diet of the lambs (Table 4). The addition of BO promoted a quadratic response to (*p* = 0.026) DM eating efficiency, with a maximal occurrence at 137 g DM intake/h at the added BO level of 12 g/kg DM total. Most of the mean time was spent idling (745.6 min/d), followed by spent ruminating (532.6 min/d) and eating (161.8 min/d). According to the LRP test, the inclusion of 0 and 12 g/kg of BO in DM was similar for DM ruminating efficiency (*p* < 0.001). The inclusion of levels above 12 g CT/kg in the diet reduced DM ruminating efficiency, but did not affect the other variables of ingestive behavior from the LRP.

### 3.3. Digestibility, N Balance and Blood Serum Metabolites

There was a linear decrease in the digestibility coefficient of DM (*p* = 0.029), EE (*p* = 0.023), and NDF_ap_ (*p* = 0.045), with increasing BO inclusion in the diet of the lambs (Table 5). There were no changes in CP (*p* = 0.33) and NFC (*p* = 0.49) digestibility in lambs with BO inclusion. Replacing ground corn with BO linearly decreased (*p* < 0.001) N intake and N balance or retention, and responded quadratically to fecal N (*p* = 0.022) and urinary N (*p* = 0.011) excretion. The basal endogenous N (BEN) was not changed (*p* = 0.33) by BO inclusion. There was no effect of BO inclusion in the diet of lambs on the serum concentrations of total protein (*p* = 0.90), albumin (*p* = 0.714), globulin (*p* = 0.64), A:G ratio (*p* = 0.63), triglyceride content (*p* = 0.15), or total cholesterol (*p* = 0.13).

From the LRP test, it was observed that the inclusion of 0 and 12 g/kg of BO in DM was similar to nutrient intake, digestibility and N balance (*p* < 0.001), indicating that the inclusion of levels above 12 g CT/kg in the diet reduced dry matter intake (DMI) and, consequently, the intake of other nutrients. Nonfibrous carbohydrates, basal endogenous nitrogen and serum metabolites did not show differences according to the LRP analysis.

## 4. Discussion

Buriti oil (BO) is largely composed of C18:1-*c-9* and monounsaturated fatty acids (MUFA) corresponding to 70% of total FA (Table 1), and this factor probably explains why DMI and digestibility were lower when the animals were fed diets supplemented with more than 12 g/kg of BO. Notably, the maximum inclusion occurred at the level of 12 g/kg BO in DM, which is lower than the values considered in the literature (up to 50 g/kg DM), which, according to Morais et al. [4], were up to 45 g BO/kg DM in goat diets. However, the composition of the oil used as well as the age of the animals change the amount of oil to be used in the ruminant diet [1,3,10,12,16,30]. In the case of the present study, the fact that BO presents MUFA-rich and young animals contributes to the LRP occurring at a lower concentration.

According to Palmquist and Mattos [10], the reduction in intake is usually caused by unsaturated fatty acid (UFA) intake and its toxicity to ruminal microorganisms, which is related to their amphiphilic nature, i.e., FAs that are soluble in both organic and water solvents are more toxic. Such acids include medium-chain FAs (from 10 to 14 carbon atoms) and long-chain MUFAs. Thus, feed intake was reduced by increasing lipid intake, and fiber digestibility (NDF), as well as the rate of passage through the gastrointestinal tract, can be both of them decreased due to the negative effect of the presence of SFAs (BO with SFA content ± 20% in DM) [4,17] on microbial growth in the rumen because the growth of cellulolytic microorganisms is particularly affected when lipids are supplied at concentrations greater than 5% in lamb diets [12,13,17]. According to Bagaldo et al. [30], a decrease in NDF digestibility can promote an energy content reduction in the diet and decrease the performance of growth.

The inclusion of BO to replace ground corn grain in the diet decreased CP intake and consequently negatively affected N intake and N balance. Despite this reduction, all experimental groups presented a positive N balance. Inclusion of levels up to 12 g/kg BO in total DM improved N utilization by reducing its fecal and urinary excretion. This improvement in N utilization may be related to the reduction in intestinal viscosity promoted by the inclusion of oil in the diet, as observed by Mir et al. [31]. This effect can lead to improved digestion and nutrient absorption in the small intestine. However, these differences in N-fecal and N-urinary excretion output did not affect the BEN in sheep, which consists of amino acids that are inevitably unavailable to the animal and are lost [32]. This result indicates that the efficiency of N use was not reduced with the inclusion of BO, probably because the N lost through feces may be associated with increased neutral detergent insoluble nitrogen (NDIN) and ADIN concentrations. This occurs in the case of adding oils to the diet but not when oilseed cakes are added to the diet of the lambs [12,33].

These changes in DM and NDF intake and digestibility promoted by partial substitution of ground corn with BO changed part of the ingestive behavior of the lambs. The lambs tended to reduce the time spent eating as they reduced the number of chews per bolus. In contrast, the chewed amount (g DM/bolus) increased. The inclusion of BO in the diet reduced the size of the mouthful, because the amount of corn grain was reduced. In addition, Lima et al. [7,15] and Bagaldo et al. [30] related that, in diets with oil inclusion, FAs are associated with hydrophobic surfaces of feed particles, which explains the low-fat toxicity when the animal is fed roughage-rich rations. In all treatments of this study, there was a large participation of roughage (400 g/kg Tifton-85 hay in DM total) in the diet. Nevertheless, there was a reduction in the intake and RE of DM and NDF.

An infinite attempt has been made to limit the extent of lipid ruminal hydrogenation and the disturbances of carbohydrate digestion by protecting this nutrient based on different techniques. This means not only the protection of lipids against microbial attack, but also the protection of microbes against the negative effect of lipids. According to Jenkins et al. [34], feeding oilseeds instead of oils provided slight protection mainly to cellulolytic bacteria [35]. According to Nagaraja et al. [36], a relevant effect of lipid supplementation is the increase in microbial protein flow, due to a reduction in protozoan concentration, resulting in greater microbial efficiency. In addition, the use of oil byproducts increases ruminal propionate production and generally reduces methanogenesis, which provides more energy to the animal [37,38].

The DMI ranged from 900 and 945 g/d (0 g/kg and 12 g/kg BO in total DM) to 740 g/d (48 g/kg BO in total DM), and was lower than that predicted by the NRC [14], which describes an intake from 1000 to 1300 g/d for the animal category used in the present study. However, all animals achieved an ADG of 200 g/d [14]. Although feeding efficiency is a dependent variable of DMI and weight gain, it increased, due to the increase in BO levels. The replacement of corn grain with BO increased the energy density (TDN) of the diets, which allowed a lower DMI. The feeding efficiency observed for the animals ingesting 36 and 48 g/kg BO in the total DM was 0.27 and 0.28, respectively, which was greater than the value suggested by the NRC [15], which reported a value of 0.25.

Despite the lambs presenting a reduction in HCW and CCW, dressing content decreased and SC increased. Dressing content (DC) is a measure of the proportion of the non-carcass components (blood, intestine, intestinal fat, and head). DC reduced from 48 to 44% when compared (0 g/kg) control to 48 g/kg of BO inclusion in lamb diets, respectively. This probably occurred due to the BO presenting a high energy density, which reduced DMI and consequently lower slaughter BW and feed volume in the Santa Ines lambs at the time of slaughter. The dressing content increases with age, mass, and fatness and is a consequence of increasing crude energy per kilogram DM in the ration, which probably explains the lower DMI, digestibility and performance in diets containing up to 12 g/kg BO inclusion [3,12,30,39]. The reduction in dressing content may also explain the lower carcass weight of the BO-fed animals [39].

Therefore, the reduction in HCW and CCW and yields was possibly due to the increase in noncarcass components, a fact related to high-fat accumulation, i.e., related to the energy density of the diet. According to Kamalzadeh et al. [40], organs and viscera have different growth rates, and are mainly influenced by the chemical composition of the diet and energy level. The Santa Ines breed accumulates large amounts of internal fat [41], a fact that may have contributed to the decrease in carcass weights and yields with increasing BO addition. SFT was not affected by the addition of BO to the diet of lambs. However, when including up to 12 g/kg of BO, the LMA of the lamb carcass was higher. The LMA is a direct response of muscle deposition, and as the animals improved the ADG up to 12 g/kg of BO, the LMA improved. The SFT recorded in the current study was 2.70 mm, and the LMA was 13.0 cm^2^, which were within the recommended ranges (LMA of 2.70 to 3.00 mm; SFT of 8 to 14 cm^2^) for lamb carcasses of animals slaughtered between 15 and 40 kg BW [3,12,39].

## 5. Conclusions

The inclusion of 12 g/kg BO in total DM of diet is recommended, because it improves the feeding intake, digestibility, and performance of lambs. However, the inclusion of BO at higher levels in the lambs’ diet is not recommended, because it can impair intake, digestibility, and animal performance. It is important to note that the use of this byproduct is recommended when it is easily available and cost effective.

## Figures and Tables

**Table 1 animals-10-01973-t001:** Chemical composition and fatty acids composition of ingredients used in lamb diets.

Item	Ingredients			
	Tifton-85 Hay	Soybean Meal	Ground Corn	Buriti Oil
Chemical Composition (g/kg DM)				
Dry matter (g/kg as fed)	872	871	872	939
Ash	60.3	60.0	16.0	-
Crude protein	35.5	443	70.4	10.6
NDIN ^1^	417	125	149	-
ADIN ^1^	142	30.3	47.5	-
Ether extract	8.0	22.2	53.5	994
Neutral detergent fiber_ap_ ^2^	729	108	115	-
Acid detergent fiber	338	67.8	48.1	
Cellulose	323	52.9	22.7	-
Hemicellulose	388	52.5	95.4	-
Acid detergent lignin	51.4	13.4	12.6	-
Nonfibrous carbohydrate	74.1	312	731	-
Fatty Acid composition (g/100 g FAME) ^3^				
Saturated fatty acids (SFA)				
C12:0	3.30	0.80	2.96	-
C14:0	1.76	0.42	1.08	0.08
C15:0	1.49	5.33	1.03	0.03
C16:0	13.5	12.3	21.8	17.7
C17:0	3.06	5.8	1.86	0.10
C18:0	1.61	7.48	5.41	1.67
Monounsaturated fatty acids (MUFA)				
C14:1	10.6	2.33	7.36	-
C15:1	6.10	1.18	5.66	-
C16:1	3.25	5.26	0.75	0.15
C17:1	3.88	3.13	0.96	-
C18:1 cis	6.88	11.7	9.43	76.2
Polyunsaturated fatty acids (PUFA)				
C18:2 cis	17.8	33.84	23.4	1.58
C18:3 n–6	1.17	0.61	6.95	-
C18:3 n–6	20.1	5.79	1.04	1.14
C20:2	0.84	0.33	1.78	-
C20:3 n-6	0.94	0.55	2.87	-
C20:3 n-6	1.58	1.43	1.78	-
C20:4	1.42	1.18	2.60	-
C20:5	0.71	0.56	1.21	-
∑SFA	24.7	32.1	34.2	19.8
∑MUFA	30.7	23.6	24.2	76.9
∑PUFA	44.1	44.3	41.7	2.72

^1^ NDIN = neutral detergent insoluble nitrogen; ADIN = acid detergent insoluble nitrogen as g/kg CP. ^2^ NDF_ap_, neutral detergent fiber corrected for ash and protein; ^3^ FAME = fatty acid methyl ester.

**Table 2 animals-10-01973-t002:** Ingredient proportion and chemical compositions of experimental lamb diets with incremental quantities of buriti oil.

Item	Buriti Oil Inclusion (g/kg DM Total)				
	0	12	24	36	48
Ingredient (g/kg DM)					
Ground corn	395	380	366	351	337
Soybean meal	190	193	195	198	200
Buriti oil	0	12	24	36	48
Mineral mixture ^1^	15.0	15.0	15.0	15.0	15.0
Tifton-85 hay	400	400	400	400	400
Chemical composition (g/kg DM)					
Dry matter (g/kg as fed)	874	874	875	876	877
Ash	56.8	56.8	56.7	56.6	56.5
Crude protein	127	127	127	127	127
Neutral detergent insoluble nitrogen ^2^	249	248	246	244	241
Acid detergent insoluble nitrogen ^2^	81.3	80.7	80.1	79.5	79.0
Ether extract	28.6	39.8	51.0	62.2	73.4
Neutral detergent fiber_ap_ ^3^	357	356	355	353	352
Acid detergent fiber	167	167	166	165	165
Nonfiber carbohydrates	378	368	358	348	339
Cellulose	148	148	148	148	147
Hemicellulose	203	202	201	199	198
Acid detergent lignin	28.1	27.9	27.8	27.6	27.5
Total digestible nutrients	698	710	723	735	747
Metabolizable energy (Mcal/kg DM)	2.52	2.56	2.61	2.66	2.70

^1^ Assurance levels (per kilogram of active elements): 120 g of calcium, 87 g of phosphorus, 147 g of sodium, 18 g of sulfur, 590 mg of copper, 40 mg of cobalt, 20 mg of chromium; 1800 mg of iron, 80 mg of iodine; 1300 mg of manganese, 15 mg of selenium; 3800 mg of zinc, 300 mg of molybdenum; maximum 870 mg of fluoride. ^2^ (g/kg CP); ^3^ NDF_ap_, corrected for ash and protein.

**Table 3 animals-10-01973-t003:** Daily nutrient intake, performance and carcass traits of grass-fed Santa Ines lambs receiving a buriti (*Mauritia flexuosa L*.) oil dietary supplement.

Item	Buriti Oil Inclusion (g/kg DM)					SEM ^1^	*p*-Value ^2^		
	0	12	24	36	48		Linear	Quadratic	LRP
Nutrient Intake (g/d)									
Dry matter	900	945	800	797	740	29.3	0.003	0.42	<0.001
Crude protein	124	118	107	104	100	5.72	0.011	0.48	<0.001
Ether extract	23.8	25.3	22.8	21.3	17.7	0.72	0.006	0.068	<0.001
Neutral detergent fiber_ap_^c3^	310	336	288	276	270	19.5	0.088	0.33	<0.001
Nonfibrous carbohydrates	285	304	241	244	213	16.2	0.017	0.63	<0.001
Total digestible nutrients	556	560	507	482	277	22.0	0.011	0.52	<0.001
ME ^4^ (Mcal/d)	2.41	2.42	2.09	2.12	2.00	0.03	<0.001	<0.001	<0.001
Performance									
Initial BW (kg)	27.6	28.1	27.9	27.9	27.9	-	-	-	-
Final BW (kg)	43.8	44.8	42.1	43.4	42.5	0.92	0.26	0.83	0.233
Total weight gain (kg)	16.2	16.7	14.2	15.5	14.6	0.65	0.13	0.53	<0.001
Average daily gain (g/d)	230	240	200	220	200	0.01	0.13	0.54	<0.001
ADG:DMI ratio (g/g)	0.25	0.25	0.25	0.28	0.27	0.01	0.012	0.18	<0.001
Carcass traits									
Slaughter BW (kg)	43.8	44.8	42.1	43.4	42.5	0.92	0.26	0.83	<0.001
Hot carcass weight (HCW, kg)	21.1	21.3	20.1	19.5	18.9	4.80	0.009	0.37	<0.001
Hot carcass yield (HCY, g/kg)	480	474	467	444	443	0.72	0.001	0.28	<0.001
Cold carcass weight (CCW, kg)	20.9	21.2	19.9	19.3	18.8	4.82	0.009	0.38	<0.001
Cold carcass yield (CCY, g/kg)	477	471	463	441	441	0.74	0.002	0.30	<0.001
Dressing content (DC, %)	48.2	47.5	47.7	44.9	44.5	0.95	0.035	0.41	<0.001
Shrink after chilling (SC, %)	0.95	0.47	1.00	1.03	0.53	<0.001	0.043	0.015	0.435
LMA ^5^ (cm^2^)	14.9	14.0	13.0	12.9	12.1	1.03	0.14	0.71	<0.001
SFT ^6^ (mm)	2.71	2.92	3.21	2.71	2.46	0.57	0.91	0.54	0.712

^1^ SEM = standard error of the mean; ^2^ Significant at *p* ≤ 0.05; LRP = linear response plateau; ^3^ NDF_ap_, corrected for ash and protein; ^4^ Metabolizable energy; ^5^
*Longissimus* muscle area; ^6^ Subcutaneous fat thickness.

**Table 4 animals-10-01973-t004:** Ingestive behavior of Santa Ines lambs fed diets with buriti (*Mauritia flexuosa L*.) oil.

Ingestive Behavior	Buriti Oil Inclusion (g/kg DM)					SEM ^1^	*p*-Value ^2^		
	0	12	24	36	48		Linear	Quadratic	LRP
Time Spent (min/d)									
Eating	179	145	164	144	177	9.29	0.051	0.38	0.542
Rumination	523	517	552	548	523	17.1	0.17	0.94	0.472
Idling	738	778	724	748	740	15.4	0.76	0.59	0.863
Chewing									
Amount (g DM/bolus)	1.70	1.90	2.30	2.60	2.10	0.20	0.002	0.82	0.434
N° chews/bolus	827	731	621	637	752	57.9	0.014	0.34	0.546
Time (min/d)	702	661	713	692	700	15.6	0.75	0.54	0.632
Efficiency (g/h)									
Eating DM	313	386	291	301	259	12.6	0.031	0.018	0.634
Eating NDF	107	137	105	105	94.0	6.51	0.19	0.026	0.843
Ruminating DM	103	110	87.1	87.8	85.4	3.64	0.0002	0.40	<0.001
Ruminating NDF	35.9	39.3	31.4	30.3	31.2	2.21	0.019	0.30	0.324

^1^ SEM = Standard error of the mean; ^2^ Significant at *p* < 0.05; LRP = linear response plateau; DM = dry matter and NDF = neutral detergent fiber.

**Table 5 animals-10-01973-t005:** Digestibility coefficient, N balance and serum metabolites of Santa Ines lambs fed diets with buriti (*Mauritia flexuosa* L.) oil.

Item	Buriti Oil Inclusion (g/kg DM)					SEM ^1^	*p*-Value ^2^		
	0	12	24	36	48		Linear	Quadratic	LRP
Digestibility Coefficient (g/1000 g intake)									
Dry matter	613	617	529	532	444	31.5	0.029	0.98	<0.001
Crude protein	592	600	600	554	446	26.0	0.34	0.32	<0.001
Ether extract	820	840	774	649	577	29.6	0.0003	0.023	<0.001
Neutral detergent fiber_ap_ ^3^	582	567	513	515	421	26.6	0.045	0.75	<0.001
Nonfibrous carbohydrates	588	620	611	547	485	42.7	0.49	0.28	0.211
Nitrogen (N) balance (g/d)									
N Intake	19.9	19.0	17.2	16.7	16.0	5.75	<0.001	0.34	<0.001
N-Urinary excretion	7.81	7.01	7.12	7.75	8.08	2.54	0.011	0.03	<0.001
N-Fecal excretion	6.49	6.15	4.75	5.00	5.21	1.66	0.022	<0.001	<0.001
N Balance	5.63	5.80	5.35	3.93	2.76	1.18	<0.001	0.18	<0.001
Basal endogenous nitrogen	5.37	5.30	5.30	5.37	5.32	0.45	0.33	0.73	0.232
Serum metabolite concentrations									
Total protein (g/dL)	7.03	7.36	6.78	7.29	7.11	0.38	0.90	0.81	0.543
Albumin (A; g/dL)	3.11	3.19	3.15	3.28	3.20	0.077	0.16	0.74	0.644
Globulin (G; g/dL)	3.92	4.17	3.63	4.01	3.91	0.089	0.64	0.78	0.843
A:G ratio	0.79	0.76	0.87	0.82	0.82	0.34	0.63	0.66	0.453
Cholesterol (mg/dL)	57.5	60.5	64.5	64.0	68.0	3.86	0.13	0.61	0.756
Triglycerides (mg/dL)	16.0	19.9	17.5	20.4	20.1	1.68	0.15	0.80	0.163

^1^ SEM = Standard error of the mean; ^2^ Significant at *p* < 0.05; LRP = Linear Response Plateau; ^3^ NDF_ap_, corrected for ash and protein.

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
