# Peer review of "Effects of the Dietary Inclusion of Buriti Oil on Lamb Performance, Carcass Traits, Digestibility, Nitrogen Balance, Ingestive Behavior and Blood Metabolites"

_animals, 2020, doi:10.3390/ani10111973_

Round 1

Reviewer 1 Report

General comments

  • The research of this manuscript is interesting and the data were adequate. However, the manuscript needs more efforts to improve its quality. You should pay more attention in the revision of Introduction and Discussion
  • In the introduction, please write several words in the relationship between Buriti oil and Santa Ines lambs. Why you select using BO in Santa Ines lambs. You have few words to interpret the lambs.
  • What are the bioactive compounds or FA composition of Buriti oil?

Line-by-line comments

Line 46: Please revise this sentence “The inclusion of BO in up to 12 g/kg total DM is recommended….”. “The inclusion of 12 g/kg BO in the total DM of diet is recommended…”

Line 51-56: I do not think this sentence has a lot meaning. suggest to delete it.

Line 78-81: Too long, please revise it.

Line 82-85: why you hypothesized that diets with a BO inclusion of up to 48 g/kg DM? why you said even if it negatively affects intake and digestibility fiber and ingestive behavior? These words might be your results, thus you could not list it in the hypothesis.

Line105: Why you select these different doses of BO? What is your reference?

Line 238: Please use the P or p consistent through the manuscript

Tables

Table 3: why the supplementation of BO decreased the Dry matter intake? You did not discuss it.

Table 5: why not statistical value for the Initial BW (kg)?

Author Response

We have thanked the adjustments suggested by the Reviewers of our paper, and we really appreciate the attention and your contribution in the analysis and correction of this paper. We have improved the paper from the Major Comments (2nd round) of the Reviewer #1, Reviewer #2 and Reviewer #3 became the paper better understandable and reading. Please accept our apologies. We have sent over again the manuscript to review the use of English and ensure that the paper flows well with appropriately constructed sentences. The changes have been highlighted in blue for checking the Editor and Reviewers. All corrections were addressed, as you can see below and in the attached file. Answers to the questions are provided below, and all the changes in the manuscript have been highlighted in yellow to Reviewer #1, green to Reviewer #2, and purple to Reviewer #3. We have understood that the changes do not guarantee the acceptance of the manuscript although we have already been grateful for them excellent collaborations.

Sincerely yours,

Leilson Rocha Bezerra

Reviewer 2 Report

The use of by-products from the processing industry in livestock feed represents an important aspect both because it allows the enhancement of products, which otherwise would represent waste, and because it allows to reduce the costs of food for livestock. The manuscript is a good contribution to the sector but before being published it needs some changes to improve it.

Specific comments
Lines 38 and 42, insert (p<0.05) to the end of the sentence
Lines 38 42 and 44, (p<0.05) (p<0.05) (p <0.001), be consistent
Line 95 "The experiment lasted 84 days and was preceded by a 14-day adaptation period." Please re-write this sentence, the experiment lasted 70 days and was preceded by a 14-day adaptation period.
Line 215 "... five treatments and eight and five experimental units for treatment, respectively." Please re-write this sentence ".... five treatments with eight and five experimental units per treatment, respectively."
Line 227 Results, consistently with the Materials and Methods, it is appropriate to report the results of the first experiment (Intake, performance and carcass traits; Ingestive behavior) and to follow those of the second experiment (digestibility trial, N balance and blood metabolites).
Line 233 Table 3, improve graphically
Line 279 (750 g / 100 g FAME), check
Line 282 ".... of BO was 12 g / kg DM, ...", check
Line 356 Conclusions, could be improved
Line 357 "Replacing up to 12 g / kg DM soybean meal with BO ...", soybean meal or ground corn? check

Author Response

We have thanked the adjustments suggested by the Reviewers of our paper, and we really appreciate the attention and your contribution in the analysis and correction of this paper. We have improved the paper from the Major Comments of the Reviewer #1, Reviewer #2 and Reviewer #3 became the paper better understandable and reading. We have sent over again the manuscript to review the use of English and ensure that the paper flows well with appropriately constructed sentences. The changes have been highlighted in blue for checking the Editor and Reviewers. All corrections were addressed, as you can see below and in the attached file. Answers to the questions are provided below, and all the changes in the manuscript have been highlighted in yellow to Reviewer #1, green to Reviewer #2, and purple to Reviewer #3. We have understood that the changes do not guarantee the acceptance of the manuscript although we have already been grateful for them excellent collaborations.

Sincerely yours,

Leilson Rocha Bezerra

Reviewer 3 Report

The paper animals-933060 entitled “Effects of the dietary inclusion of buriti oil on lamb performance, carcass traits, digestibility, nitrogen balance, ingestive behavior and blood metabolites” aimed at understanding the role of dietary inclusion of the byproduct buriti oil in the diet of lambs for meat production. The paper was well written and the methods were well conducted. Before the acceptance of the paper, some minor comments that needs to be addressed are listed below:

L62: Substitute with adverb “independently”.

L122: Please substitute the term “various” with “incremental”.

L138: Please delete “again” and add “in order to”… determine.

L187-188: Change into “immediately after feeding and after 4 h”.

L219: Which is the meaning of “adjusted”? Please specify the statistical method used, as an example “All statistical analyses were performed on the adjusted initial BW using it as covariate” or similar.

Table 5: I suggest to add the acronyms in the table for CCW, HCY, CCY and many others and define them on the bottom of the table as for SEM or, if the authors prefer, it can be used the parenthesis, as an example Hot carcass weight (HCW, kg). This could help the reader.

Discussion section: Please add a brief introduction before discussing data.

L337-339: Please rewrite this sentence more clearly.

L339: Please change “Bo” into “BO”.

Conclusion section: Please add a brief introduction.

Author Response

We have thanked the adjustments suggested by the Reviewers of our paper, and we really appreciate the attention and your contribution in the analysis and correction of this paper. We have improved the paper from the Major Comments (2nd round) of the Reviewer #1, Reviewer #2 and Reviewer #3 became the paper better understandable and reading. Please accept our apologies. We have sent over again the manuscript to review the use of English and ensure that the paper flows well with appropriately constructed sentences. The changes have been highlighted in blue for checking the Editor and Reviewers. All corrections were addressed, as you can see below and in the attached file. Answers to the questions are provided below, and all the changes in the manuscript have been highlighted in yellow Reviewer #1, green to Reviewer #2, and purple to Reviewer #3. We have understood that the changes do not guarantee the acceptance of the manuscript although we have already been grateful for them excellent collaborations.

Sincerely yours,

Leilson Rocha Bezerra

Round 2

Reviewer 1 Report

Accept

Reviewer 2 Report

The authors provide good and sufficient data with well written. The article can be accepted.

This manuscript is a resubmission of an earlier submission. The following is a list of the peer review reports and author responses from that submission.

Round 1

Reviewer 1 Report

The study “Effects of the dietary inclusion of buriti oil on lamb performance, carcass traits, digestibility, nitrogen balance, ingestive behavior and blood metabolites” provided basic data for the application of buriti oil in the lamb diet. But it needs further improvement for publication. Comments and suggestions as following:

L23: “reduces” or “induces”?

L41: The expression of “quadratically increased” is unscientific. Linearly increase or decrease is good. But for quadratic effect, the expression should be like … quadratically response to …

L100: The control was one of you treatment, so the treatment number is five not four.

L101: Table 1 and Table 2 should put in the part of “Experimental diets and management”.

L126-127and L271-274: the expression of DP and SC should be the same as that in table5. Please check the unit of SC in table 5.

L132: Please supplement the replicate number of lamb for ingestive behavior.

L145: If you take this part as the second Exp., then you should point out that the treatment and management of this diet. How long does the Exp. 2 last, only 28 days?

L171: The blood samples were collected on day 4 and 6, what’s the results indicated in the paper-the average values or …?

L199-201: The metabolizable energy value is calculated value or measured value?

L211: Replace the “four treatments” by “five treatments”

L228-230: The P value of BO supplementation effects should be added in table 3 and other tables. The data of energy intake and digestibility should be supplemented in table 3. The BEN was calculated in this table, so it is helpful to explain the results if you can give the real N-Balance value.

L271-273: In table 5, please check the data of Shrink after chilling. The SEM value larger than all the average values, which means the data deviation is too large, you may need use the transformation analysis.

L276: 750 g/100 g ??? Supplement the reference of the information.

L284-293: In this part, the author need provided the SFA concentration in the diet or in the SB if the author try to explain the results from the SFA aspect.

L291: The sentence need rewrite or cite new reference, because the lipid content only in the last group greater than 7%.

L320: Please provide more information about the “two groups”.

L320-321: Remove this sentence “other factors, such as…”, this sentence has no relation with you research.

L322-324: This statement contradicts the results in Table 3.

L399-340: The description not in accordance with the result in table 5.

L364-365: Please remove the sentence-“the weight reduction occurred…”. This is only the conjecture, and there is no exp. Data to prove it.

L364-365: There is no reference value if the recommendation is BO in the diet can up to 48 g/kg DM.

Author Response

The letter with corrections and answers is attached.

Reviewer 2 Report

These studies evaluated the addition of varying amounts of Buriti oil to feed rations for ram lamb growth and some carcass traits.  The authors determined that 12g/kg was the optimal addition of Buriti oil for Santa Ines ram lambs.  Buriti oil could be a lower cost feed additive in areas that produce the oil.

More information is needed about the animals:

Specifically were they related?  If so, how (e.g. paternal half sibs, all from 1 farm that has 3 sires)?  If there was relatedness this should be included in the statistical model (for example sire as a random effect).  

For your randomized block design ram lamb is the experimental unit not a replicate.  You have 8 experimental units per treatment in study 1 and 5 experimental units per treatment in study 2.  A replicate would be if you had 16 experimental units per treatment divided into two different times (8 EU for November and December and 8 EU for December and January).  Remove replicate from the methods when referencing experimental units (ram lambs).  

There is some variablility in the ram lambs starting weight where lambs in the 12g/kg group are slightly heavier that the other groups.  This must be accounted for in the statistical model as a covariate or subtracting the start weight from the subsequent weights to get a true gain per treatment value.

Author Response

(The authors gave the same response as above.)
